# Explaining a Deep Reinforcement Learning Docking Agent Using Linear Model Trees with User Adapted Visualization

**Vilde B. Gjærum** [1,*], **Inga Strümke** [1], **Ole Andreas Alsos** [2] **and Anastasios M. Lekkas** [1,3]

1   Department of Engineering Cybernetics, Norwegian University of Science and Technology, 7034 Trondheim, Norway; inga.strumke@ntnu.no (I.S.); anastasios.lekkas@ntnu.no (A.M.L.)
2   Department of Design, Norwegian University of Science and Technology, 7491 Trondheim, Norway; oleanda@ntnu.no
3   Centre for Autonomous Marine Operations and Systems, Norwegian University of Science and Technology, 7052 Trondheim, Norway
*   Correspondence: vilde.gjarum@ntnu.no

**Abstract:** Deep neural networks (DNNs) can be useful within the marine robotics field, but their utility value is restricted by their black-box nature. Explainable artificial intelligence methods attempt to understand how such black-boxes make their decisions. In this work, linear model trees (LMTs) are used to approximate the DNN controlling an autonomous surface vessel (ASV) in a simulated environment and then run in parallel with the DNN to give explanations in the form of feature attributions in real-time. How well a model can be understood depends not only on the explanation itself, but also on how well it is presented and adapted to the receiver of said explanation. Different end-users may need both different types of explanations, as well as different representations of these. The main contributions of this work are (1) significantly improving both the accuracy and the build time of a greedy approach for building LMTs by introducing ordering of features in the splitting of the tree, (2) giving an overview of the characteristics of the seafarer/operator and the developer as two different end-users of the agent and receiver of the explanations, and (3) suggesting a visualization of the docking agent, the environment, and the feature attributions given by the LMT for when the developer is the end-user of the system, and another visualization for when the seafarer or operator is the end-user, based on their different characteristics.

**Keywords:** deep reinforcement learning; autonomous surface vessel; explainable artificial intelligence; linear model trees

## 1. Introduction

Machine learning is the sub-field of artificial intelligence (AI) dedicated to self-learning systems that use data to adjust their predictions. Among the most remarkable advancements in machine learning methods has been the evolution from artificial neural networks to deep architectures, known as deep neural networks (DNNs), forming the class of deep learning [1,2]. reinforcement learning (RL) is a branch of machine learning where an agent learns a strategy, referred to as a policy, which the agent uses to interact with an environment based on an evaluation of the agent's interactions with the environment [3], called rewards. That is, the policy maps from a state to an action, similar to a controller. Several noteworthy accomplishments have been made with the use of deep reinforcement learning (DRL), such as learning to play Atari games directly from image pixels [4] or discovering new strategies in a simulated hide-and-seek environment [5]. DRL has also shown to be a very useful tool for accomplishing difficult tasks in robotics, one advantage being that it does not require a mathematical model of the agent or the environment. In [6], DRL was used to perform 20 different simulated physical tasks. In [7], DRL was used to conduct various manipulation tasks with a dexterous, robotic hand. In [8], a DRL-agent learned to walk on flat surfaces, but could also handle unseen, more challenging sur-

faces. The potential for reduced costs and increased safety has inspired the work towards autonomous ships, and the industry has already shown promising autonomous surface vessels (ASVs), such as Falco [9] and Yara Birkeland [10]. DRL has also been used for marine operations, such as autonomous path-following [11–13], collision avoidance [14,15], and for docking [16,17]. In [17], a DRL-agent was trained to perform docking of an ASV in a simulated environment based on Trondheim harbor. Both the agent and the simulator from [17] is used here and are described in Section 2, Appendices A and B. Even though the agent showed promising and rather convincing results, its applicability to real-life problems is reduced by the lack of understanding of how the DNN-policy makes its decisions. This is because the many parameters and interconnections of DNNs make their inner workings hard for humans to interpret. For this reason, DNNs are considered to be *black-boxes*. The field of explainable artificial intelligence (XAI) is dedicated to developing methods for explaining such black-box models [18]. The objective is to gain an increased understanding of how the black-box model works and why it behaves the way it does. XAI methods can thus be used to interpret and justify the decisions made by a black-box model, control and prevent erroneous actions, improve the model, and even discover new strategies, correlations in the data set or application [19]. In [20], the importance of explaining AI-systems to non-expert users is highlighted, especially with consideration for the preparation for wider-scale operations of ASVs. The combination of explanations and thorough testing of the AI system is crucial for gaining the trust needed for the autonomous system to be deployed [21]. The authors of [21] also argue that depending on the role and needs of the recipient of the explanation, the explanations should be customized regarding several aspects:

1. Whether all the details of an explanation should be provided, or if it is preferable to highlight only the most relevant parts, with respect to the specific end-user, of the explanation;
2. Whether the end-user need to process the explanations in real-time or not;
3. In what way the explanation will be presented to the end-user.

In this work, we consider two different end-users, the developer and the seafarer/operator. Their main differences lies in their background knowledge, how much risk they associate with the predictions made by the DNN, and how fast they need to evaluate the predictions from the DNN and the explanations from the explainer.

Among the most widely used explanation methods are local interpretable model-agnostic explanations (LIME) [22], Anchors [23], integrated gradients (IG) [24], Shapley additive explanations (SHAP) [25], and SAGE [26]. The main characteristics of XAI-methods are outlined in Table 1. IG is a model-specific method, only applicable to differentiable models, while LIME, Anchors, SHAP, and Shapley additive global importance (SAGE) are model-agnostic methods. While SAGE provides global explanations, SHAP, LIME, Anchors, and IG give *local explanations*. Preliminary work was presented in [27], where we approximated the DRL-policy from [17] with a linear model tree (LMT) and used the linear functions in the active leaf node to form explanations in the form of feature attributions. The LMTs was built by a greedy method, which was quite sensitive to the dataset. To remedy this, a very time-demanding iterative data sampling process was used to ensure that the LMT got enough samples from regions it did not approximate as well. In this paper, we improve the approximation by enforcing the order of which features are used when searching for the splits in each branch node at different depths of the tree to better match the guidance system logic. Not only does this speed up the time it takes to build one tree, but the iterative data sampling process was deemed unnecessary when ordered feature splitting was used. Additionally, two different visualizations for two different end-users with regards to their characteristics and needs are suggested. Following the main characteristics of XAI-methods, LMTs is a post-hoc, model-agnostic explanation method giving local explanations in the form of feature attributions. Even though the feature attributions are used to form the local explanations in this work, it should be noted that instead of creating an explanation model for specific data points, the LMT approximates

the full model across its whole range of validity. So, the explanations provided by the LMT are local, but since decision trees (DTs) are considered interpretable, in theory, the LMT also yields a global explanation of the full model. However, note that for all practical means, the global interpretability of a DT is reduced quickly as the size of the tree increases. The LMT is intended to run in real-time, parallel to the full model, to provide explanations in the form of feature attributions for its predictions. The ability of LMTs to run real-time combined with their inherent transparency are the two main benefits of using them to explain black-box models used in robotic applications such as docking. The terms used in relation to LMTs and DRL are described in Table 2.

Our main contributions are the following:

- An improved and faster building process of LMTs from [27] by introducing re-ordering to the splitting sequence of the input features, to better match the way guidance systems work. This made the iterative data sampling process slowing down the building process from [27] unnecessary;
- An overview of the background knowledge, skills, needs, and requirements the different end-users of the docking agents have;
- Two different visualizations of the explanations based on the characteristics of each end-user.

**Table 1.** Main characteristics of XAI-methods.

| | | |
|---|---|---|
| **Scope of explanation** | Local/ Global | The scope of the explanations range from local explanations, where only one instance is explained, to global, where the entire model is explained. This is not a binary category, as groups of similar instances can be explained at the same time. |
| **Complexity of model to be explained** | Intrinsic/ Post-hoc | Models that are self-explanatory, such as simple linear regression, are called intrinsically explainable models. More complex methods, such as most DNNs or other models considered to be black-boxes however, cannot be easily understood by humans, so a post-hoc XAI-method must be applied to the model to aid with the understanding of it. |
| **Applicability of XAI-method** | Model-agnostic/ Model-specific | A model-agnostic XAI-method treats the model to be explained as a black-box, that is the XAI-method only cares about the inputs and outputs of the model to be explained. Thus, it can be applied to any model. A model-specific XAI-method, as the name implies, can only be applied to one specific model. |

**Table 2.** Description of terms used in relation with the LMT and DRL.

| LMT | DRL | Description |
|---|---|---|
| Input features | States | The information that the model is trained and later used to predict on, in this case, a description of the environment as provided to the policy and the policy approximator, given in Equation (2). For this application, the states are describing how the vessel is situated in the harbor. |
| Predictions | Actions | The model output, given in Equation (1). For this application, the actions are directly controlling the force and angle of the vessel's thrusters. |
| Policy approximator | Policy | The model itself, providing a mapping from input features to predictions or states to actions, respectively. The policy corresponds to the *controller* in robotics, while the LMT acts as the policy approximator and is only used to generate explanations. |
| Explainer | Agent | The application of the model. In the current setting, the agent comprises the policy and the vessel, while the explanations are formed using the LMT to generate feature attributions and visualizations. |

## 2. Preliminaries

In this section, the DRL agent, as well as the training environment used for its development, are presented. For more details regarding the docking agent, the reader is referred to Appendix A. For more details regarding the simulated environment, the reader is referred to Appendix B. Both appendices are summarizing work done in [17].

### 2.1. The ASV Docking Problem

Docking is the process of taking a vessel from being in open waters to being fastened to a specific location along the quay, referred to as the *berthing point*. The process can be divided into the following three stages:

1. The approach phase;
2. The berthing phase;
3. The mooring phase.

During the approach phase, the vessel moves from open seas to confined waters. In the berthing phase, the vessel maneuvers inside confined waters until it is parked at a location close to the berthing point. In the mooring phase, the vessel is fastened to the berthing point. Docking is considered to be a challenging task since it requires complex decision-making and significant fine-tuning of actions. In addition to being difficult to model, external disturbances affect the vessel more at low speeds than at high speeds. Thus, their impact on the movement of the vessel increases when the vessel operates at low speeds close to obstacles. The simulation environment used in this work is based on Trondheim harbor and is the same as the one used in [17]. Figure 1a shows a snapshot of the simulation environment. An illustration of the vessel is shown in Figure 1b. The vessel has three thrusters: a tunnel thruster in the front and two azimuth thrusters at the back. The vessel is controlled using the control inputs

$$\mathbf{A} = [f_1, f_2, f_3, \alpha_1, \alpha_2], \tag{1}$$

where $f_1, f_2$ are restricted to the range $[-70 \text{ kN}, 100 \text{ kN}]$ and $\alpha_1, \alpha_2$ to the range $[-90 \text{ degrees}, 90 \text{ degrees}]$ represent the force and the angle of the two azimuth thrusters, respectively. The tunnel thruster is controlled by changing its force, $f_3$, in the range $[-50 \text{ kN}, 50 \text{ kN}]$.

The features representing the vessel's state and relative position in the environment form the vector

$$\mathbf{x} = [\tilde{x}, \tilde{y}, \tilde{\psi}, u, v, r, l, d_{obs}, \tilde{\psi}_{obs}].\tag{2}$$

Here, $\tilde{x}$ and $\tilde{y}$ represent the relative distance to the berthing point in the vessel's body frame, in which $u, v, r$ represent the vessel's velocity. The variable $\tilde{\psi}$ represents the difference between the actual heading and the heading desired at the berthing point. Note that since $\tilde{x}$ and $\tilde{y}$ are in body frame, they are only aligned with the environment axis if $\tilde{\psi}$ is zero and aligned with the environment frames. That is, $\tilde{x}$ is not necessarily in the south-north direction, and $\tilde{y}$ is not necessarily in the west-east direction. The binary variable $l$ indicates whether or not the vessel has made contact (i.e., collided) with an obstacle, which, as discussed in Section 2.2, is mainly used during training. The vessel's position relative to the closest point of the closest obstacle is described by the two variables d$d_{obs}$ and $\tilde{\psi}_{obs}$s. The direction of where the obstacle is in relation to the vessels own heading in body frame is given by $\tilde{\psi}_{obs}$s, while the distance along this direction is given by $d_{obs}$.

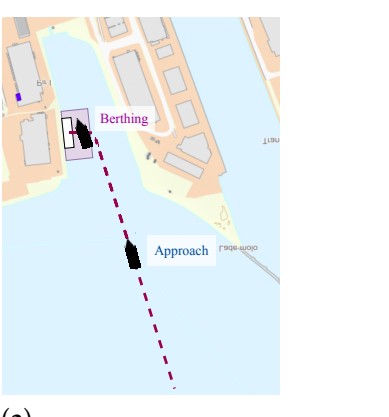

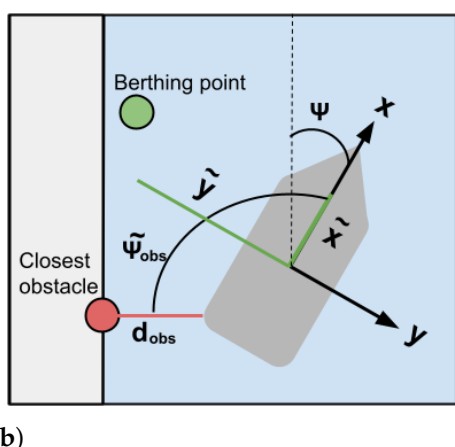

(**a**)          (**b**)

**Figure 1.** (**a**) The simulation environment, and (**b**) an illustration of the vessel's states.

Even though the DNN should be able to understand the necessary dynamics of the vessel based on the vessel's pose and velocities, states that give information more directly, which is important for the docking problem, were used in addition, because practice shows that this gives quicker and more stable training. A restriction in the simulated environment is that the agent is not allowed to make any contact with the harbor under any circumstances, although gentle contact with the harbor under low speeds while berthing is usually allowed in real life. The binary variable $l$ is only used during training of the DRL-policy through giving a large penalty and ending the episode.

### 2.2. The Docking Agent

As previously discussed, the problem of docking is challenging due to several reasons, one being that it is hard to achieve adequate mathematical models of all the aspects affecting the operation, which is crucial of most traditional control methods. In [17], an RL-agent learned how to perform berthing from substantial distances—up to 400 m from the berthing point, corresponding to LP4:Distance berthing in [17]—without using any additional models of the environment or vessel. RL is the branch of machine learning dedicated to learning by interacting with the environment and receiving rewards for different states based on their desirability. The reward function is chosen or designed by the programmer and is crucial for the learning process of the agent. Extensive work was done in engineering a fitting reward function for the task in this paper, and the objectives for the RL-agent are the following:

1. Avoiding any obstacles, specifically keeping $d_{obs} > 0$;

2.  Reaching and staying at the berthing point, specifically achieve a stable situation with $\tilde{x} = \tilde{y} = \tilde{\psi} = 0$.

Note that COLREG (Convention on the International Regulations for Preventing Collisions at Sea) (https://www.imo.org/en/About/Conventions/Pages/COLREG.aspx, accessed on 26 September 2021) is not taken into consideration. These objectives are given to the RL-agent through the following reward function

$$R(\tilde{x}, \tilde{y}, l, d_{obs}) = R_d + R_{\tilde{\psi}} + R_{obs} + R_{\dot{d}}. \tag{3}$$

Here, $R_d$ rewards the agent for minimizing the distance to the berthing point. Given that the distance to the berthing point is small enough, rewards for achieving the desired heading are given through $R_{\tilde{\psi}}$. The agent was given significant negative rewards for getting close to, and especially, making contact with any obstacles through $R_{obs}$ since this is of high priority. However, this made the agent hesitant to get close to the berthing point since this is very close to the harbor, which in the agent's point of view is an obstacle. Therefore, the reward component $R_{\dot{d}}$, which rewards decreasing the distance to the berthing point, was designed.

To train the DRL-agent, the proximal policy optimization (PPO) algorithm from [28] was used. It is a *stochastic, on policy* algorithm that uses a trust-region to prevent too large updates to the policy based on a training batch, which can lead to getting trapped in a local minimum. The trust-region is the area in which the approximation of the gradient descent of the policy is accurate. To prevent the training to become too constrained to this trust-region, the trust-region does not set hard boundaries for the exploration but is rather included in the objective by giving penalties to the updates that encourage not leaving the trust-region. The resulting PPO-trained neural network has two hidden layers, consisting of 400 neurons each. The hidden layer's nodes use the rectified linear units (ReLu) activation function [29], while the output layer uses the hyperbolic tangent function, restricting the outputs to the range $[-1, 1]$. The agent converged after approximately 6 million interactions, i.e., instances of having a state, performing an action, and receiving a reward. One episode consists of maximum 2500 interactions(or steps) given that the vessel does not collide with the harbor limits. Thus, it took at least 2400 episodes, but most likely more since the agent is expected to perform poorly early in the training process.

## 3. Linear Model Trees

decision trees (DTs) form a class of machine learning algorithms based on conditional control statements, and are capable of solving many classification and regression problems. Their main advantages are being both easily visualized and interpretable for humans. A DT consists of branch and leaf nodes, where the branch nodes perform data splitting based on the control statements, and the leaf nodes perform the DTs prediction. In its simplest form, a DT has univariate splits, i.e., it splits based on only one feature at a time, and each leaf node has a constant prediction. *Oblique DTs* have multivariate splits in the branch nodes, making them less interpretable and significantly increasing their building time, due to which they are not used in this work. *Model trees* are DTs where the constant predictions in the leaf nodes are replaced by a prediction model, for example, a linear regression model or a DNN, so that the tree maps the input to the appropriate model. The simplest version of model trees are *linear model trees (LMTs)*, which have a linear function in the leaf nodes. As illustrated in Figure 2, an LMT makes out a piecewise linear function and the number of regions resulting from the splits of the tree correspond to the number of leaf nodes.

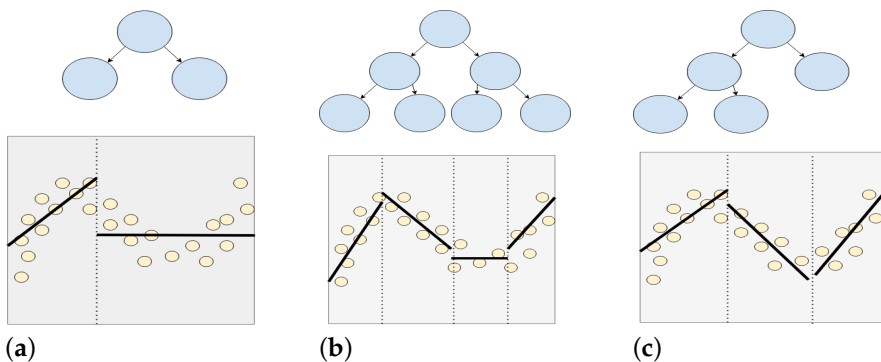

**Figure 2.** Illustrations of how two perfect LMTs of depth (**a**) two, (**b**) three, and (**c**) an imperfect LMT of depth three, fit to the same data set. The number of regions estimated by a linear function correspond to the number of leaf nodes.

The problem of building an LMT for a data set $(X, Y)$ can be expressed as

$$\min_{a,t,w} = \sum_{\forall (x,y) \in (X,Y)} (y - f(x))^2,$$ (4)

where $f(x)$ is the prediction made by the LMT, which we express as follows

$$f(x) = \sum_{l \in \forall \text{ leaf nodes}} f_l(x) \prod_{n \in l^{la}} \{a_n^T x < t_n\} \prod_{n \in l^{ra}} \{a_n^T x \geq t_n\}.$$ (5)

Here, $a_n$ is a standard basis vector in the chosen coordinate basis, which is in our case that of the vessel, while $t_n$ is the *threshold* value upon which node $n$ is split. A leaf node $l$'s ascendants are its left and right ascendants, $l^{la}$ and $l^{ra}$. The linear function $f_l$ in leaf node $l$, is given by

$$f_l(x) = \sum_{f=1}^{F} (w_f x_f) + w_{F+1},$$ (6)

where $F$ is the number of input features, i.e., states.

It is sometimes stated that DTs are fully interpretable models, but this is an oversimplification for most practical means. As outlined in [30], transparency can be understood as simulatable, decomposable, or algorithmic transparency. To be simulatable transparent, the model as a whole must be simple enough that a human can easily interpret it. This also implies that both the input features and the output features must be easily understandable. Provided that the inputs and outputs are understandable and that the trees have a reasonable size, both DTs and LMTs are simulatable transparent. To be decomposable transparent, all parts of the model must be simulatable transparent. This means that an LMT that is too big to be simulatable transparent, is still decomposable transparent, since all its parts, i.e., its subtrees, are still simulatable transparent. Finally, algorithmic transparent methods are those that can be analyzed using mathematical tools. Thus, LMTs are always decomposable and algorithmic transparent, and whether they are also simulatable transparent depends on the size of the specific tree.

### 3.1. Heuristic Tree Building

The LMTs used here are constructed by a modified version of Algorithm 1 from [27]. Since building an optimal DT given a data set $\mathcal{D}$ is an NP-complete problem [31], our approach is heuristic, which is common for most approaches to building DTs (see e.g., CART [32], ID3 [33], and C4.5 [34]).

---

**Algorithm 1:** The LMT algorithm from [27].

---

**Require:**
 *Training data $\mathcal{D}$*
 *Maximum number of leaf nodes $N$*
 *Minimum number of data samples for leaf nodes $M$*
 **while** *number of leaf nodes is less than $N$* **do**
  **if** *there exist a node that fulfills all splitting criteria* **then**
   Choose node to split
   Perform splitting
   Calculate best potential split for the newly created nodes
  **else**
   **return** root node
  **end**
 **end**

---

A so-called *perfect* DT is a tree with binary splits where all the leaf nodes have the same depth. The trees in Figure 2a,b are examples of perfect DTs. However, as pointed out in [35], perfect DTs are often unnecessarily big. Consider the docking problem, the complexity of the agent's behavior will vary in different parts of the harbor and with different positions and velocities. For example, it is expected that the maneuvers required close to the berthing point will be more intricate than at open seas. If the DT is to be perfect, it will either not be deep enough to approximate the behavior close to the berthing point, or it may overfit to the regions that require less complex behavior, such as for open seas. For the same reason, the stopping criteria were changed from maximum depth to maximum number of leaf nodes, which allows the DT to grow deeper in areas that require more splits, resulting in an imperfect tree. Figure 2b,c illustrates the difference between an imperfect DT and a perfect DT. One way of searching for splitting conditions for a node is to order the values for each feature, and try threshold values in the middle between two neighboring feature values. However, for large data sets, this procedure is very computationally expensive. Therefore, a search grid evenly distributed from the lowest to the highest feature values is used to find the split thresholds. The splitting condition for a node is found via

$$\mathcal{F}, t_n = \underset{\mathcal{F},n}{\mathrm{argmin}}(loss(\mathcal{D}_L) + loss(\mathcal{D}_R)) \,, \tag{7}$$

where $\mathcal{F}$ and $t_n$ are the feature and threshold, respectively, the node is to split upon, given the data samples in $\mathcal{D}$. That is, the non-zero entry of the basis vector $a_n$ for node $n$ in Equation (5) corresponds to $\mathcal{F}$. The data sets $\mathcal{D}_L$ and $\mathcal{D}_R$ are subsets of $\mathcal{D}$, and result from the split of a node. Each branch node splits the data it receives into a left and right part, so all data points end up in exactly one leaf node. Each node splits the data it receives according to

$$\begin{aligned}
\mathcal{D}_L &= x \in \mathcal{D} \quad \text{if} \quad x_{\mathcal{F}} \leq t_n \,, \\
\mathcal{D}_R &= x \in \mathcal{D} \quad \text{if} \quad x_{\mathcal{F}} > t_n \,,
\end{aligned} \tag{8}$$

where $x_{\mathcal{F}}$ is the data sample $x$'s value for feature $\mathcal{F}$. Since not all possible thresholds are explored, and there is no guarantee for global optimality since this method is greedy, there is no need for the algorithm to be deterministic and yield exactly the same tree in each run. Instead, having the process include some randomness leads to a wider exploration in the same runtime if run in parallel. The $n$'th threshold is

$$t_n = min(\mathcal{D}^{\mathcal{F}}) + (n+r)\frac{(max(\mathcal{D}^{\mathcal{F}}) - min(\mathcal{D}^{\mathcal{F}}))}{N} \,, \tag{9}$$

where $\mathcal{D}^{\mathcal{F}}$ are all the values of feature $\mathcal{F}$ in the data set $\mathcal{D}$, $N$ is the number of thresholds in the grid search, and $r$ is a random number that alters the threshold value in the range $\pm 2\%$. The next node $n_s$ to split is chosen using

$$n_s = \underset{n}{\text{argmax}}((1+r)(loss(\mathcal{D}_L^n) + loss(\mathcal{D}_R^n))), \qquad (10)$$

where $\mathcal{D}_L^n$ and $\mathcal{D}_R^n$ are the losses of the left and right child nodes, respectively, of node $n$, given its best split variables $\mathcal{F}$ and $t_n$. The linear functions showed in Equation (6) in the leaf nodes are calculated by performing ordinary least squares regression on the data $\mathcal{D}_l$ belonging to leaf node $l$.

As our aim is for the LMT to be a faithful explanation model for the DRL model, the loss in Equation (10) is calculated as the mean squared error (MSE) between the prediction of the DRL model and that of the linear function fitted by linear regression in the leaf nodes.

When tested, Algorithm 1 turned out to be very sensitive to the data set $\mathcal{D}$, which is a well-known problem for DTs. How many data points are needed to represent an area properly, depends on how complex the DRL model is in that area.

To mitigate this, we performed the data sampling and tree building iteratively, according to the following algorithm

This process was repeated for 10 iterations, before checking which resulting LMT performed best on an independent validation set. The best chosen LMT tree was built on the ninth iteration.

In [27], an imperfect LMT with a total of 681 leaf nodes was trained to approximate and serve as an explanation model for the DRL-model presented in Section 2.2. The tree model is inarguably too large to be considered simulatable transparent, but it can still be used to map the input features, i.e., the states, to the predictions, i.e., the actions. Furthermore, sub-parts of the tree are still considered simulatable transparent. The maximum depth of the tree was 15, while the shallowest leaf node was at depth 5. As the vessel has five control inputs, the DRL model has five outputs, and so must the LMT. This can be achieved either by building one LMT for each control input or by building one LMT for all the control inputs. In the latter case, every leaf node contains a fitted linear function for each of the control inputs, and the average loss is used when fitting and evaluating the splits. Consequently, this approach requires the control inputs, respectively the LMT outputs, to be normalized. The latter approach was used both in [27] and the present work, because simplicity is desired, and because it is much more time-demanding to build five trees instead of just one.

### 3.2. Building Linear Model Trees Utilizing Ordered Feature Splitting

Although the LMT used in [27] did show promising results, there are two important drawbacks. Primarily, the process of building it is slow because several trees must be built, and data sampling has to be done for several iterations. This leads to a larger data set, which again increases the time it takes to build an LMT for each iteration. Secondly, the resulting tree is very large and thus, as mentioned, in no way simulatable transparent, which is a significant drawback since the LMT is used as an explanatory model.

To address this, we set the order in which the LMT building process searches for feature splits. This is done by letting the LMT search for splits on the following features, and in the following order:

1. $\tilde{x}, \tilde{y}, \tilde{\psi}$;
2. $d_{obs}, \tilde{\psi}_{obs}$;
3. $u, v, r$.

As mentioned, the binary variable $l$ is only for penalizing the DRL-agent during training and ending the episode, and it will therefore not be used for the LMTs. The order is set to better match guidance system logic, however, a more systematic approach remains for future work. During training, the criteria for a split to be valid is that the overall loss decreases, and that the child node receives a minimum number of data samples, here **M**.

Once these criteria are met, the node is split. If the criteria are not met after trying all features in the three feature groups, the tree stops growing. How these criteria are set is essential for the building process of the LMT, and both the complexity of the problem and the size of the dataset must be taken into consideration when setting the criteria. If the criteria are set too strict by requiring either too big of a loss decrease or by setting the minimum number of samples in each leaf node too high, the tree might underfit and not be able to represent important aspects of the problem. If the criteria are set not strict enough, the tree might overfit to the dataset.

As expected, this approach to searching the features for splitting decreases the time needed to train the LMT, since the number of feature and threshold pairs are reduced in the split search. Additionally, with ordered feature splits the iterative data sampling process shown in Algorithm 2 was deemed unnecessary. More importantly, the resulting tree is more interpretable for humans since similar features are close to each other, making it easier to locate which parts of the tree are relevant in different situations.

---

**Algorithm 2:** The data sampling process from [27].

> **Require:**
>  Maximum number of iterations ***Max_it***
> Maximum number of leaf nodes ***N***
> Minimum number of data samples for leaf nodes ***M***
> it = 0
> **while** *number of iterations it is less than **Max_it*** **do**
> > $\mathcal{D}_{it+1} \leftarrow$ sample_from_environment($LMT_{it}$)
> > $LMT_{it} \leftarrow$ Algorithm 1($\mathcal{D}_{it}$,**M,N**)
> > it++
>
> **end**

---

## 4. Increasing Model Interpretability Using Linear Model Trees

The goal of approximating the DRL model with an LMT is to use the inherent transparency of the LMT and its intuitive structure to efficiently obtain an importance ranking of the input features, i.e., the states of the vessel. However, as previously emphasized, although DTs, and consequently LMTs, are transparent, this does not necessarily make them easily understandable for humans. In this section, we first discuss how the linear functions in the leaf nodes can be used to obtain explanations for a prediction in the form of feature attributions. Next, we demonstrate how these feature attributions can be visualized together with the environment as well as the states and actions of the vessel to obtain a more comprehensible picture.

### 4.1. Extracting Feature Attributions from the Leaf Nodes

LMTs can give local explanations in the form of feature attributions, which can be seen as giving credit or blame to the input features for the output, in essence, feature attributions are answering the question "how much did each input feature affect the model's output?". The local explanations are calculated utilizing the coefficient in the linear function in the leaf nodes and the values of the instance to be explained. The linear functions in the leaf nodes take the form of and Equation (11) shows how the importance for the feature $\mathcal{F}$, $I_{\mathcal{F}}$ is calculated.

$$I_{\mathcal{F}} = \frac{w_{\mathcal{F}} x_{\mathcal{F}}}{\sum_{f \in \forall \mathcal{F}} |w_f x_f|}, \tag{11}$$

where $w_{\mathcal{F}}$ and $x_{\mathcal{F}}$ is the coefficient from the linear function in the leaf node in Equation (5) and the sample's value for feature $\mathcal{F}$. It should be noted that the constant coefficient $w_F + 1$ from Equation (5) is not included in Equation (11), which means that if the linear function in a leaf node is a constant, no feature attributions can be calculated. Additionally, it should be noted that when forming these local explanations, only the function in the leaf node is taken into consideration, even though the path from the root node to this leaf node is not

irrelevant and most likely should be considered. However, including the paths in (both local and global) explanations should not be done carelessly since even irreducible DTs can have irrelevant splits [36].

### 4.2. Visualization of Feature Attributions

Different users require both different explanations and different representations of the explanations, states, and actions. For this work, two users of the black-box model are considered, namely the developer and the seafarer/operator. The developer wants to use the XAI-method to verify that the black-box model works as intended, detect edge cases or erroneous behavior to improve the model, as well as understanding how the model behaves. On the other hand, the seafarer/operator uses the XAI-method as a supporting tool to monitor and control the autonomous agent's behavior to assess whether or not they should intervene to prevent a dangerous situation or accident. An important difference is that the operator/seafarer has personal risks associated with the potential erroneous behavior of the model, whereas the developer has not. The different relation to the black-box model, the environment, and the XAI-system for the two users is shown in Figure 3. Where the developer can carefully inspect the model's behavior in a simulated environment with no time pressure, the seafarer/operator must make assessments within a short time span with risk of serious consequences for vessel, crew, and equipment. Additionally, the seafarer/operator has a lot of other sources of information, both from other sensors and displays, but also from their own senses. The main differences that need to be taken into account when deciding how to convey the explanations to the specific user are outlined in Table 3.

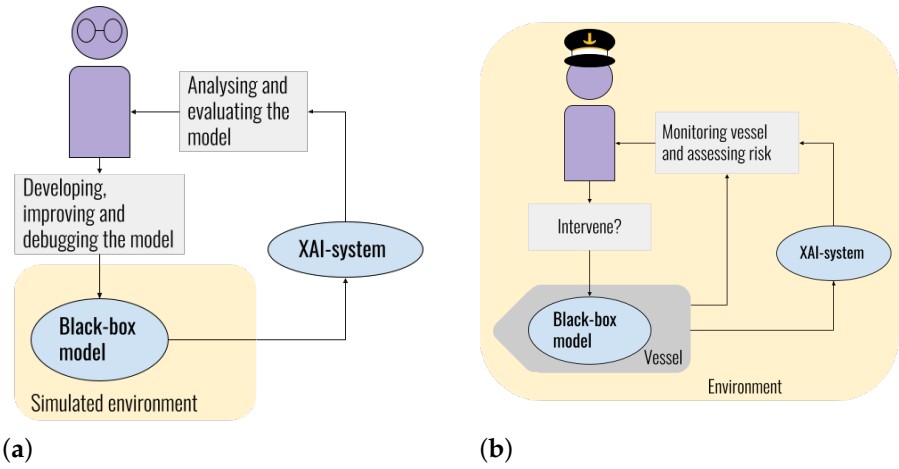

(**a**)                                                                                                     (**b**)

**Figure 3.** Illustration of (**a**) the developer's and (**b**) the operator/seafarer's different relations to the agent and the environment.

**Table 3.** Differences between developer and operator/seafarer.

|  | Developer | Operator/Seafarer |
|---|---|---|
| **Background knowledge** | Good analytical skills, but not necessarily domain knowledge | Domain knowledge, but not necessarily good analytical skills |
| **Environment** | Works in simulated environments or digital twins without risk of physical damage | Works with the physical vessel, with risks for material damage and potentially personnel injury |
| **Risk** | Works with a risk-free simulated environment | Works in a physical environment where errors can compromise safety of units involved |

**Table 3.** *Cont.*

|  | Developer | Operator/Seafarer |
|---|---|---|
| **Urgency** | Analyzes the model offline with no time pressure | Monitors the controller via the XAI module real-time under time pressure |
| **Tools** | Has access to analytical and mathematical tools | Has no analytical or mathematical tools available |
| **Information design** | Prefers information enabling thorough and analytic investigation of the controller's behavior | Prefers information suitable for fast processing, and related to the vessel |
| **Level of detail** | Desires high level of detail, has low risk of cognitive overload as information originates from one source only and the working environment is stress-free | Only interested in the necessary information, having several sources of information and a potentially stressful working environment, creating a risk for cognitive overload |
| **Event frequency** | Interested in examining the controller's behavior over the entire state space | Not interested in experiencing states that might lead to undesired behavior or dangerous situations |
| **Edge cases** | Uses edge cases to detect undesirable or unexpected behavior | Does not wish to experience edge cases that involve higher risk of faulty controller behavior |
| **Intervention** | Does not intervene if undesirable or unexpected behavior is discovered | Intervenes if entering or experiencing state that lead to undesired behavior to avoid dangerous situations |

To aid the developer in thoroughly investigating the step-by-step state-action pairs with their corresponding feature attributions the plots in Figure 4 are suggested. In Figure 4a, the feature attributions are plotted for each step. The feature attributions should be studied together with the state and action plots of Figure 4b,c. Figure 4 contains a lot of information that requires a lot of time to analyze, so this type of visualization cannot be used in real-time. For a user like the operator/seafarer, another type of representation of this information is needed. One aspect that makes it hard for humans, and even domain experts such as operators/seafarers, to process the information is the fact that the vessel has nine state features and five control inputs. Additionally, $f_1$ and $\alpha_1$, and $f_2$ and $\alpha_2$ are controlling the same motors, and it is not possible to understand the agent's behavior as a whole while looking at cooperating actions independently. To remedy this, the actions are mapped to and visualized on the vessel for faster comprehension, as can be seen Figure 5. Additionally, feature attributions for the five actions are combined as follows

$$I^{\mathcal{F}} = \sum_{a \in \mathcal{A}} |I_a^{\mathcal{F}}| \tag{12}$$

where $I^{\mathcal{F}}$ is the overall importance for the feature $\mathcal{F}$. Still, having to consider feature attributions for nine features is too much to take into consideration in a stressful environment with time pressure, so the feature attributions are further compressed, as shown in Table 4. It is important to note that the feature importance is not confused with the actual values of the features. High importance for the velocity does not mean that the vessel has a high velocity, it just means that the velocity played an important part when the action was predicted. The pipeline between the DRL-agent, the LMT and the visualization tools, and the end-users after the DNN is trained and the LMT is built is shown in Figure 6.

**Table 4.** Overview of mapping from features as the agent and explainer receives them to the compressed feature representation used for the visualizations for the operator/seafarer.

| Compressed Features | Features | Compressed Feature Importance |
|---|---|---|
| **Distance to berthing point** | $\tilde{x}, \tilde{y}$ | $I^D = I^{\tilde{x}} + I^{\tilde{y}}$ |
| **Velocity** | $u, v, r$ | $I^V = I^u + I^v + I^r$ |
| **Obstacle** | $d_{obs}, \tilde{\psi}_{obs}$ | $I^O = I^{d_{obs}} + I^{\tilde{\psi}_{obs}}$ |
| **Heading** | $\tilde{\psi}$ | $I^H = I^{\tilde{\psi}}$ |

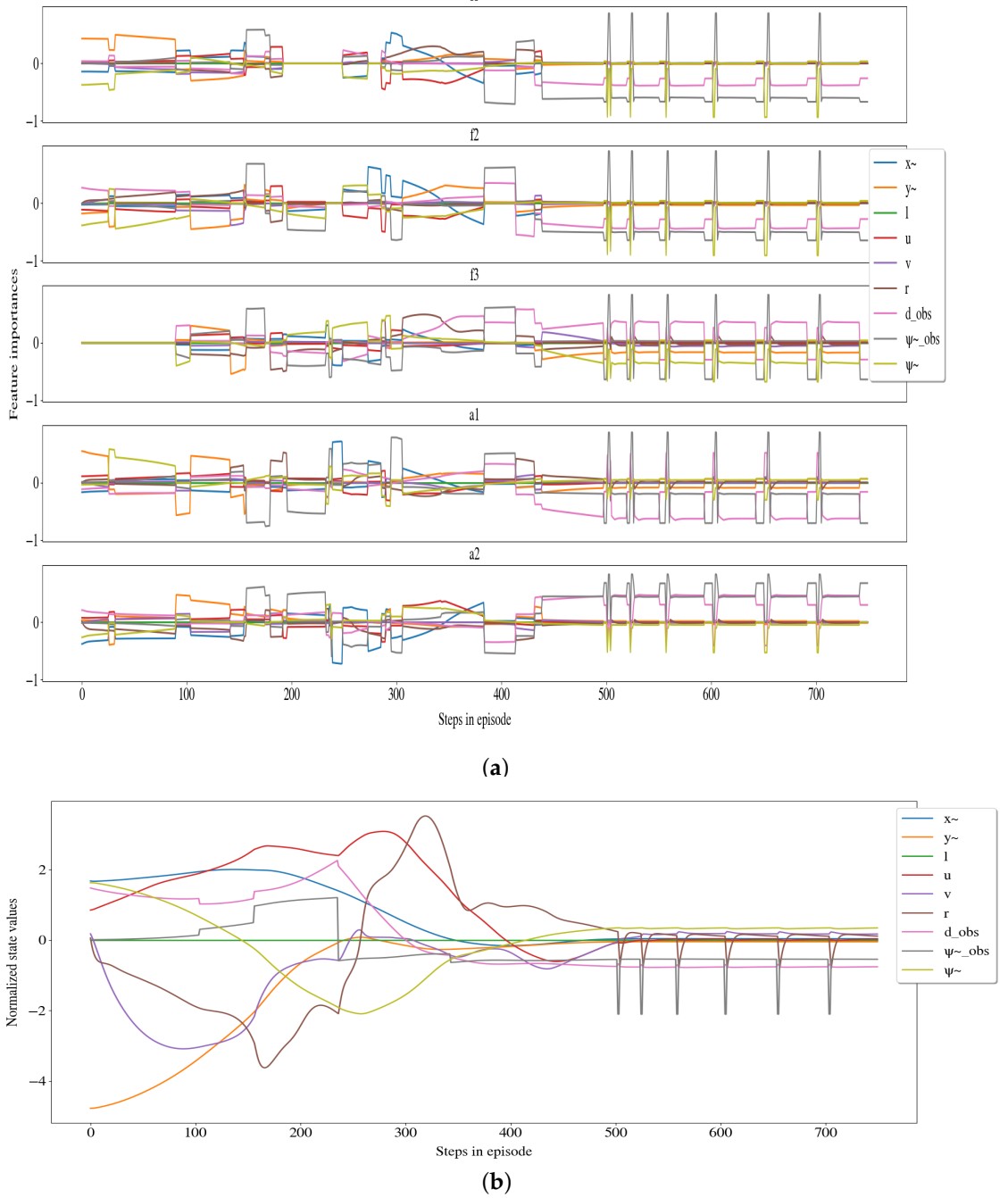

(**a**)

(**b**)

**Figure 4.** *Cont.*

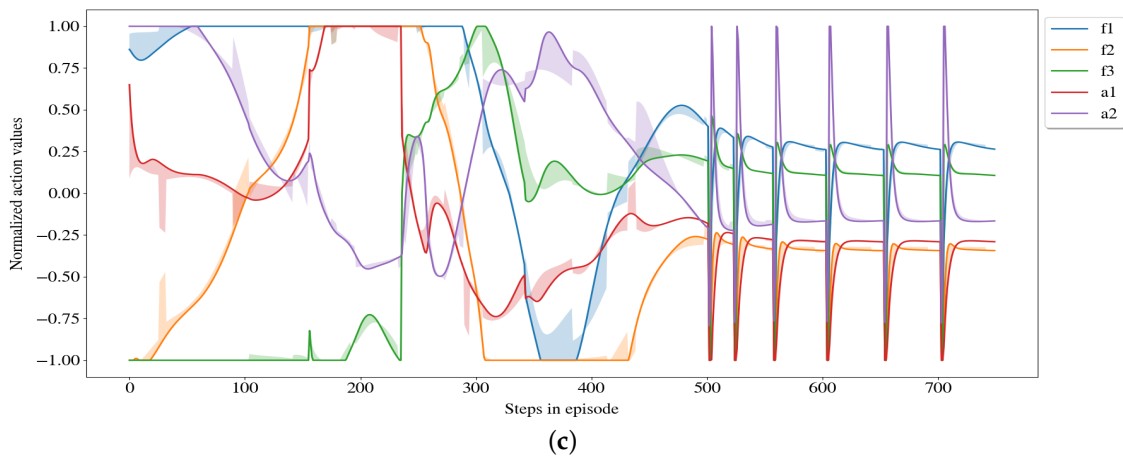

**(c)**

**Figure 4.** The visualization of the (**a**) feature attributions, (**b**) states, and (**c**) actions and from one episode for the developer. The shaded area in the action-plot shows the difference between the actions taken by the DRL-agent and predicted by the LMT.

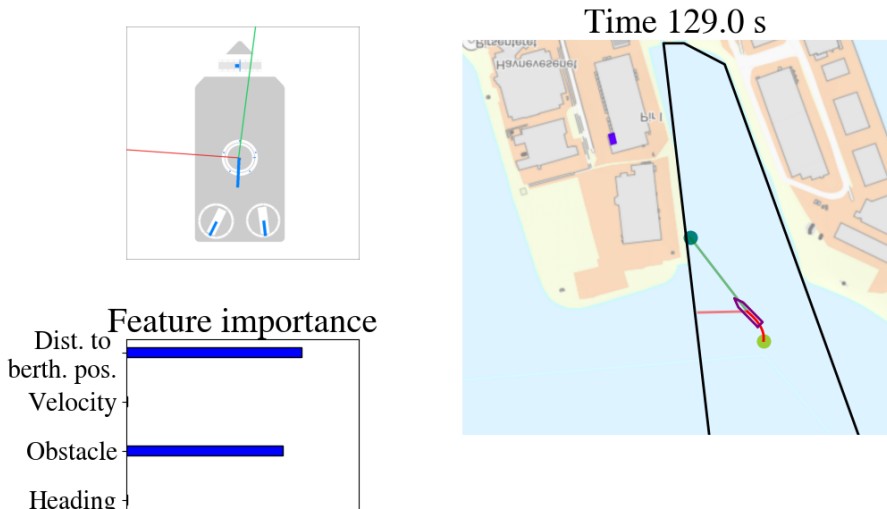

**Figure 5.** The visualization of the environment, vessel, and compressed feature attributions for the seafarer/operator. The top left part of the figure shows both the states and the actions on the vessel, along with the total forces and moment acting on the vessel. The compressed feature *Distance to berthing position* is shortened to *Dist. to berth. pos.*

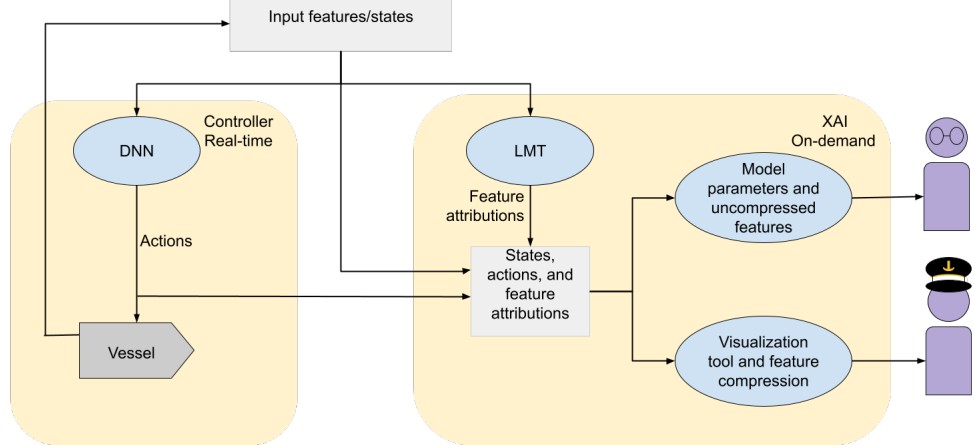

**Figure 6.** Pipeline after both the LMT and DNN are trained.

## 5. Results

To create the data set to build the LMTs, 1000 unique starting points were found, whereas 800 of these were used as starting points for the training set, 50 for the validation set, and the remaining 150 for the test set. The complete data sets consisted of data from runs performed by the RL-agent with these starting points. In this chapter the LMT process presented in Section 3.1 and the LMT process utilizing ordered feature splitting presented in Section 3.2 will be evaluated and compared.

### 5.1. Structure of Linear Model Trees

There is an important difference between building *the optimal tree given a data set*, and building *the optimal tree given a specific structure of the tree and a given data set.* In this work, we take on the problem of building an LMT given a maximum number of leaf nodes, univariate, binary splits, and a given data set. In Table 5, the structures of the two best LMTs built using ordered feature splitting and one LMT built by the purely greedy approach. The trees built with ordered feature splitting resulted in smaller trees than when the trees were built without the ordered feature splitting.

**Table 5.** Overview of the structures of the different LMTs. LMT OFS denotes LMTs where ordered feature splitting was utilized, while the number denotes the total number of leaf nodes the LMT has.

| Name of LMT | Number of Leaf Nodes | Depth of Deepest Node(s) | Depth of Shallowest Node(s) |
|---|---|---|---|
| LMT 467 | 467 | 16 | 3 |
| LMT OFS 100 | 100 | 11 | 3 |
| LMT OFS 312 | 312 | 12 | 3 |

### 5.2. Computational Complexity

To compare the computational complexity of building LMTs with the algorithm presented in Section 3.1 and the version that limits the number of features considered at each split as presented in Section 3.2 the time it takes to build trees with 10 and 50 leaf nodes are compared. The different run times are presented in Table 6. LMT OFS is significantly faster than LMT, and the difference (naturally) increases when the size of the trees increases.

**Table 6.** Run time for the different algorithms for trees of different sizes.

| Algorithm | Build Time for 10 Leaf Nodes | Build Time for 50 Leaf Nodes |
|---|---|---|
| LMT | 74.75 s | 171.45 s |
| LMT+ OFS | 52.748 s | 117.91 s |

### 5.3. Evaluating the Fidelity

The most important aspect when choosing which tree to use is how well the tree approximated the DRL-agent, i.e., the fidelity, which will be evaluated based on the following metrics:

1.  The average error between the DRL-agent's output and the tree's output given the same input state as presented in Section 5.3.1;
2.  The trees' path when running the vessel in the simulator compared to the path taken by the DRL-agent as presented in Section 5.3.2;
3.  The error between the resulting forces and moment based on the predicted actions as presented in Section 5.3.3;
4.  The rewards of the PPO-policy's and the LMT OFS 312's given the same starting points are compared in Section 5.4.

### 5.3.1. Output Error

The mean absolute error and the standard deviation can be seen in Table 7. Both the LMT OFS 100 and LMT OFS 312 has better accuracy and precision than LMT 467, despite LMT 467 being significantly larger. LMT OFS 312 also has better accuracy and precision than LMT OFS 100 on all actions. Additionally, using ordered feature splitting gave consistently better results than without, and the building process became less sensitive to the dataset.

**Table 7.** Output error analysis for the three different LMTs LMT OFS 100, LMT OFS 312, and LMT 467. The improvements from the LMT presented in [27] are highlighted in red.

| LMT OFS 100 | | |
|---|---|---|
| **Output feature** | **Mean absolute error** | **Error standard deviation** |
| $f_1$ (kN) | 4.57 (2.68%) | 9.31 (5.5%) |
| $f_2$ (kN) | 4.018 (2.36%) | 7.261 (4.2%) |
| $\alpha_1$ (deg) | 3.43 (1.9%) | 7.66 (4.3%) |
| $\alpha_2$ (deg) | 4.81 (2.67%) | 8.04 (4.4%) |
| $f_3$ (kN) | 1.77 (1.77%) | 4.049 (4.05%) |
| **LMT OFS 312** | | |
| **Output feature** | **Mean absolute error** | **Error standard deviation** |
| $f_1$ (kN) | 3.55 (2.08%) (−7.22%) | 7.78 (4.57%) (−7.27%) |
| $f_2$ (kN) | 3.33 (1.95%) (−6.35%) | 7.085 (4.16%) (−5.675%) |
| $\alpha_1$ (deg) | 2.463 (1.36%) (−7.84%) | 6.93 (3.85%) (−6.12%) |
| $\alpha_2$ (deg) | 3.66 (2.15%) (−5,48%) | 8.03 (4.45%) (−3.42%) |
| $f_3$ (kN) | 1.302 (1.3%) (−7.78%) | 3.513 (3.51%) (−12.387%) |
| **LMT 467** | | |
| **Output feature** | **Mean absolute error** | **Error standard deviation** |
| $f_1$ ( kN ) | 11.85 (6.97%) | 19.07 (11.22%) |
| $f_2$ ( kN ) | 9.039 (5.32%) | 17.38 (10.22%) |
| $\alpha_1$ ( deg ) | 7.9 (4.3%) | 14.32 (7.96%) |
| $\alpha_2$ ( deg ) | 14.09 (7.83%) | 18.91 (10.51%) |
| $f_3$ ( kN ) | 3.84 (3.84%) | 6.83 (6.83%) |

### 5.3.2. Comparing the Paths of the Agent and of the Linear Model Trees

If the LMT has approximated the PPO-policy well enough, the LMT should be able to replicate the PPO-policy's behavior. Therefore, one way of evaluating how well the LMT has approximated the PPO-policy is to compare the paths of their runs given the same starting point. Plots for the four agents from four different starting points can be seen in Figures 7 and 8. Figure 7 shows a difficult scenario where the agent must first steer the vessel backwards followed by straightening the yaw while simultaneously controlling the surge and sway. Unlike Figure 7, Figure 8 does not require a turn, but the path is close to the boundaries and deviations from this path will quickly lead to the vessel making contact with the harbor limits. The DRL-agent's behavior is shown in Figures 7a and 8a, the LMT OFS 100's behavior in Figures 7b and 8b, the LMT OFS 312's behavior in Figures 7c and 8c, and finally the LMT 467's behavior in Figures 7d and 8d. In the episode shown in Figure 8 it is clear that the LMT OFS 312 performs best out of the three. In the episode shown in Figure 7 only LMT DK 100 and LMT OFS 312 complete the episode, while LMT 467 makes

contact with the harbor limits while attempting the last part of the docking. LMT OFS 312 mimics the behavior of the DRL-agent better than LMT OFS 100, as can be seen in Figure 7.

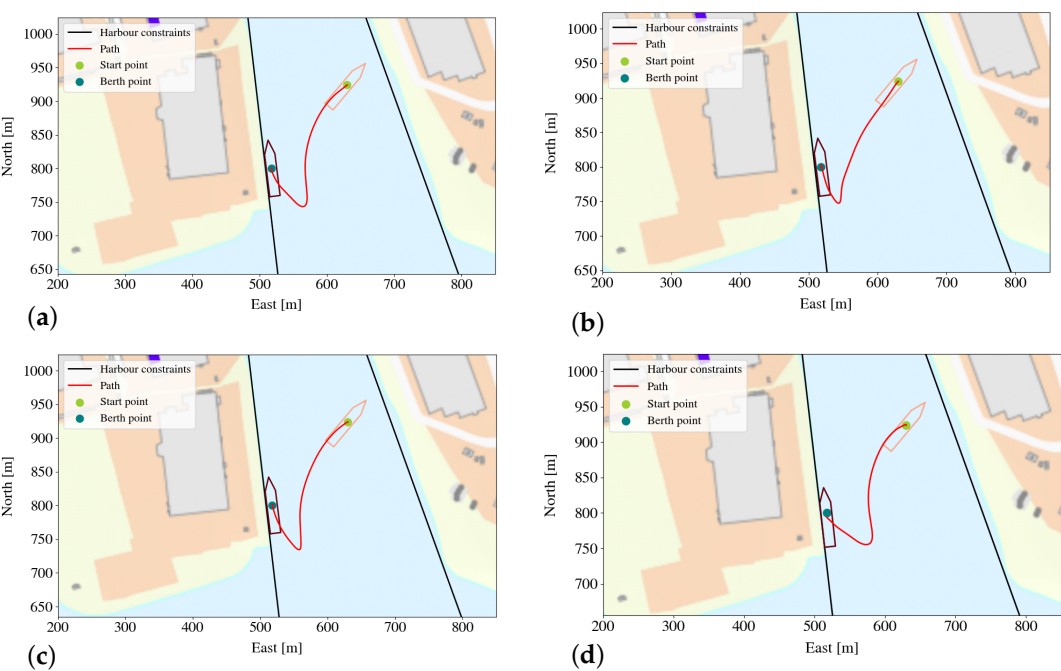

**Figure 7.** (**a**) Successful run by the PPO-policy, (**b**) failed run by the LMT OFS 100 , (**c**) failed run by the LMT OFS 312 , and (**d**) failed run by the LMT 467 .

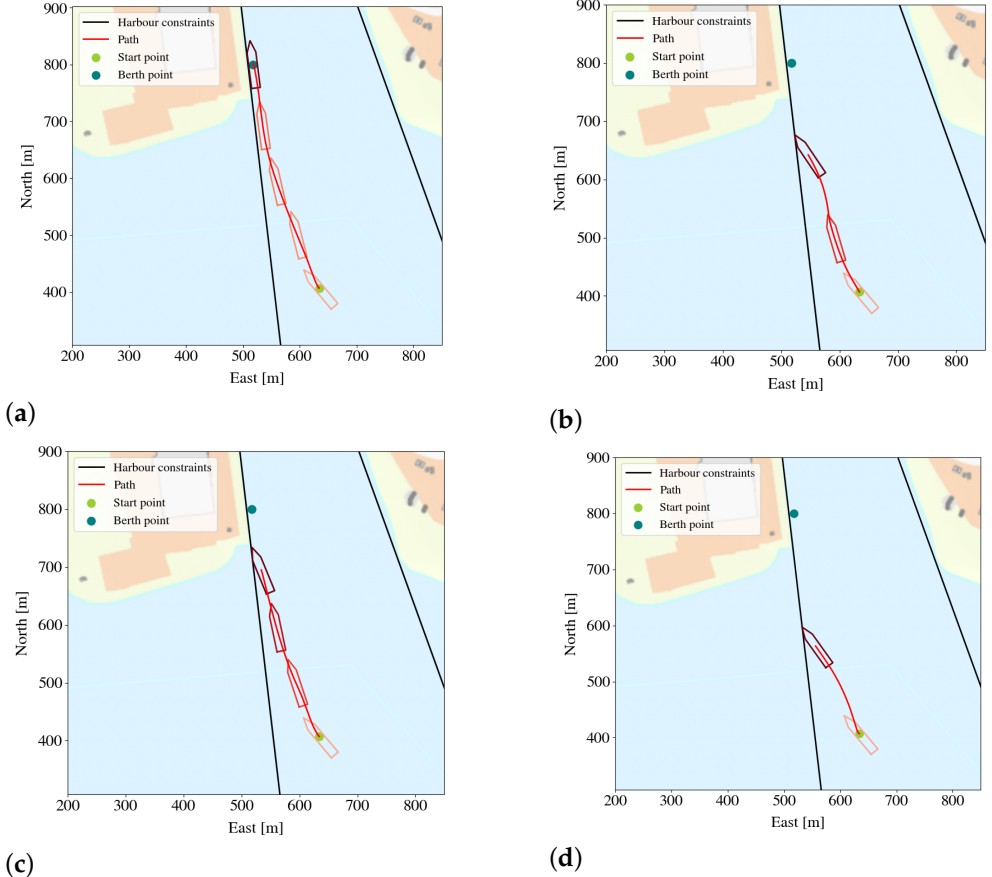

**Figure 8.** (**a**) Successful run by the PPO-policy, (**b**) failed run by the LMT OFS 100, (**c**) failed run by the LMT OFS 312 , and (**d**) failed run by the LMT 467.

### 5.3.3. Comparison of Resulting Forces and Moment on Vessel

To further investigate the behavior of the policy and the LMT, we look at the forces acting on the vessel that result from the actions taken. This is because there are many combinations of actions that may result in the same overall forces. This also means that small deviations in each action may accumulate, causing the policy and the LMT to predict very different forces, despite their action predictions being similar. On the other hand, if the force of a thruster is zero then its angle does not matter, but it may still look like an important error. We calculate the overall forces predicted by the two models as

$$F_x = \sum_{i=1}^{3} f_i cos(\alpha_i), \tag{13}$$

$$F_y = \sum_{i=1}^{3} f_i sin(\alpha_i), \tag{14}$$

$$T = \sum_{i=1}^{3} f_i \left( l_{i_x} sin(\alpha_i) - l_{i_y} cos(\alpha_i) \right), \tag{15}$$

where $F_x$ denotes the applied force in the *x*-direction, $F_y$ the applied force in the *y*-direction, and $T$ the applied torque, all three in the body frame. The forces' arms of moment are given by $l_{i_x}$ and $l_{i_y}$. Figures 9 and 10 show the forces and moments predicted by both the PPO-policy and the LMT. The LMT predictions do not follow the PPO-policy forces and moment perfectly, but the behavior is very similar. Note that, as was also the case for the actions and feature attributions, the actions predicted by the LMT sometimes change abruptly. This happens when there is a change in which leaf node in the tree is being used to make the prediction.

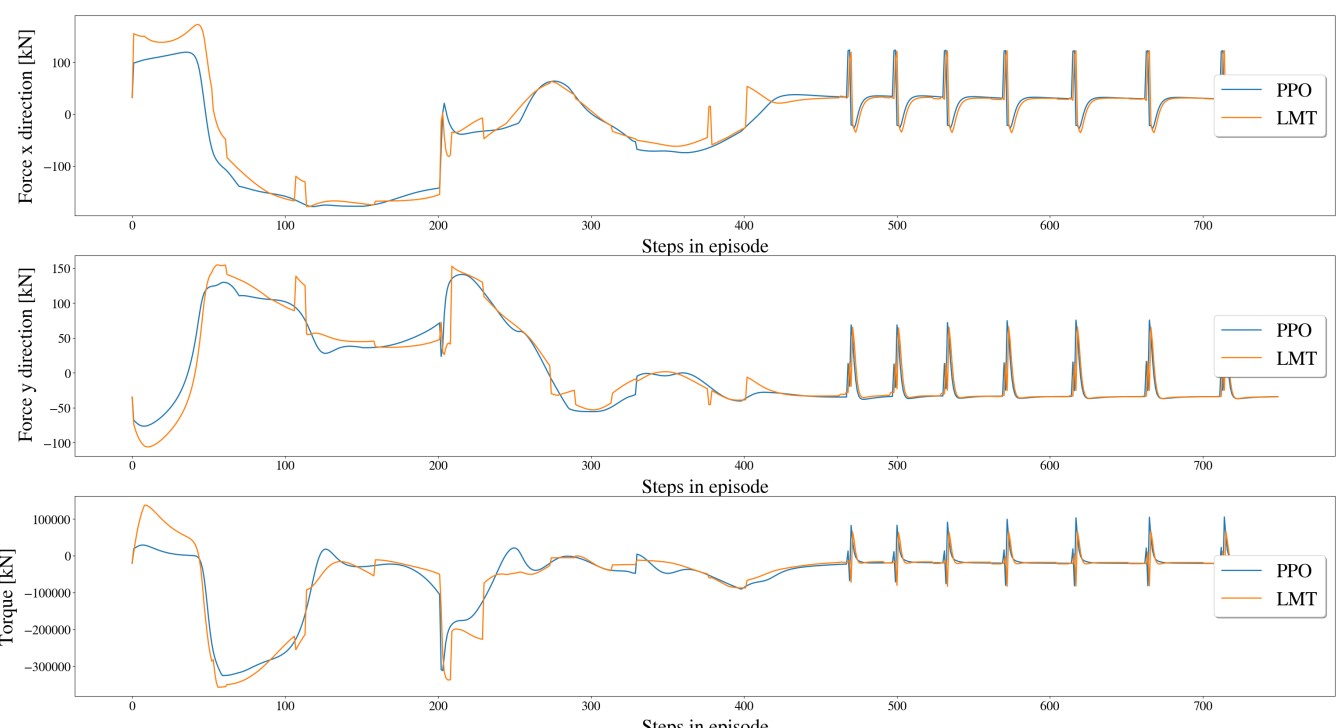

**Figure 9.** Plot of total force and moment predicted by both the LMT and the PPO-policy for an episode where the PPO-policy actions are given to the vessel.

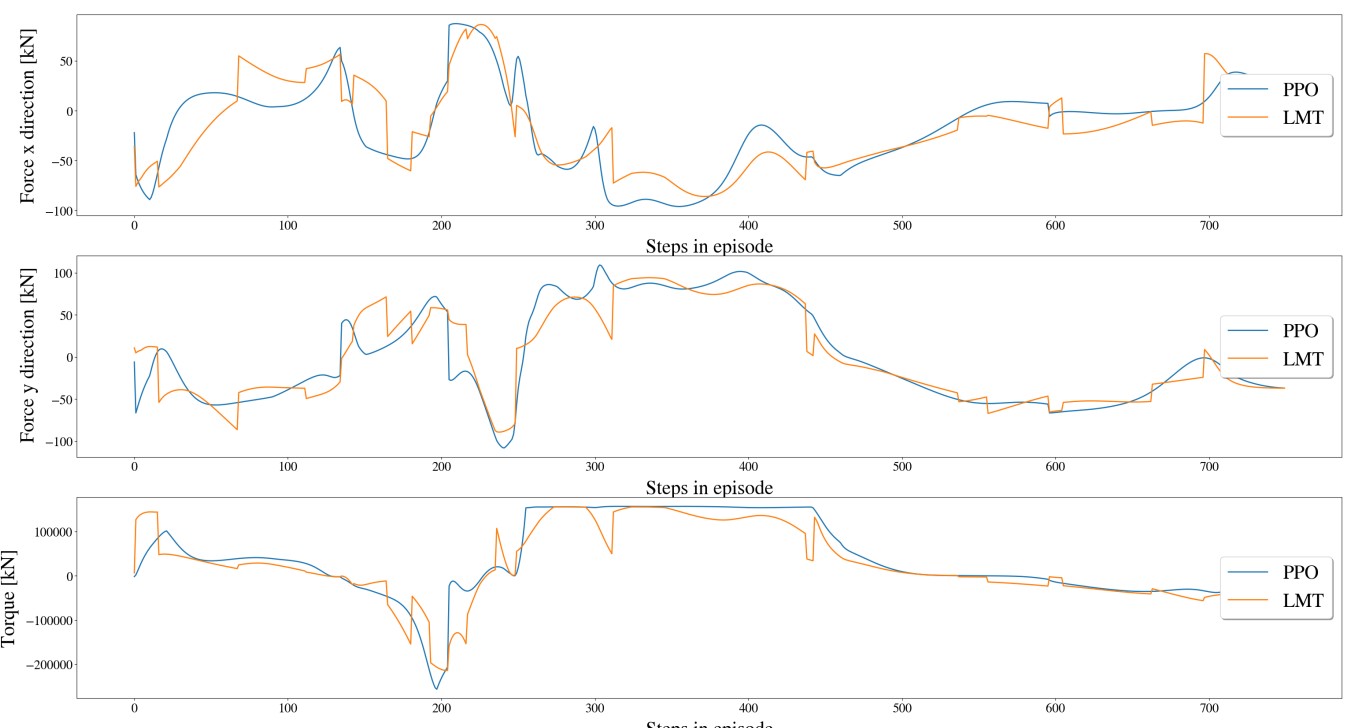

**Figure 10.** Plot of total force and moment predicted by both the LMT and the PPO-policy for an episode where the PPO-policy actions are given to the vessel.

### 5.4. Comparison of Rewards

The LMT is trained without any knowledge of the reward function, whereas the DRL-agent's training relies heavily on it. However, it is expected that the LMT receives approximately the same rewards as the DRL-agent throughout an episode since they should behave similarly. In Figure 11, an episode where the LMT and PPO-policy behaves very similarly and their corresponding rewards can be seen. Since they have such similar paths their rewards are also similar, though with small deviations. The docking problem is a complex problem with many possible solutions, and thus, many different reward functions ought to lead to a viable solution. An example of two different paths successfully leading to the berthing point from the same starting point can be seen in Figure 12. Even though both the LMT and the PPO-policy successfully bring the vessel to the berthing point, the PPO-policy receives a higher cumulative reward. In cases like this, where the LMT ends up taking a different, but still viable, path than the PPO-policy does, the output error will be high. It might be interesting to evaluate the LMT in the same way as the PPO-policy, because if the LMT behaves as well as the PPO-policy, the LMT could replace the PPO-policy entirely, which would be beneficial because then we would be certain that the feature attributions would be completely correct. Nevertheless,as could be seen in Figure 8, the PPO-policy performs better than the LMTs.

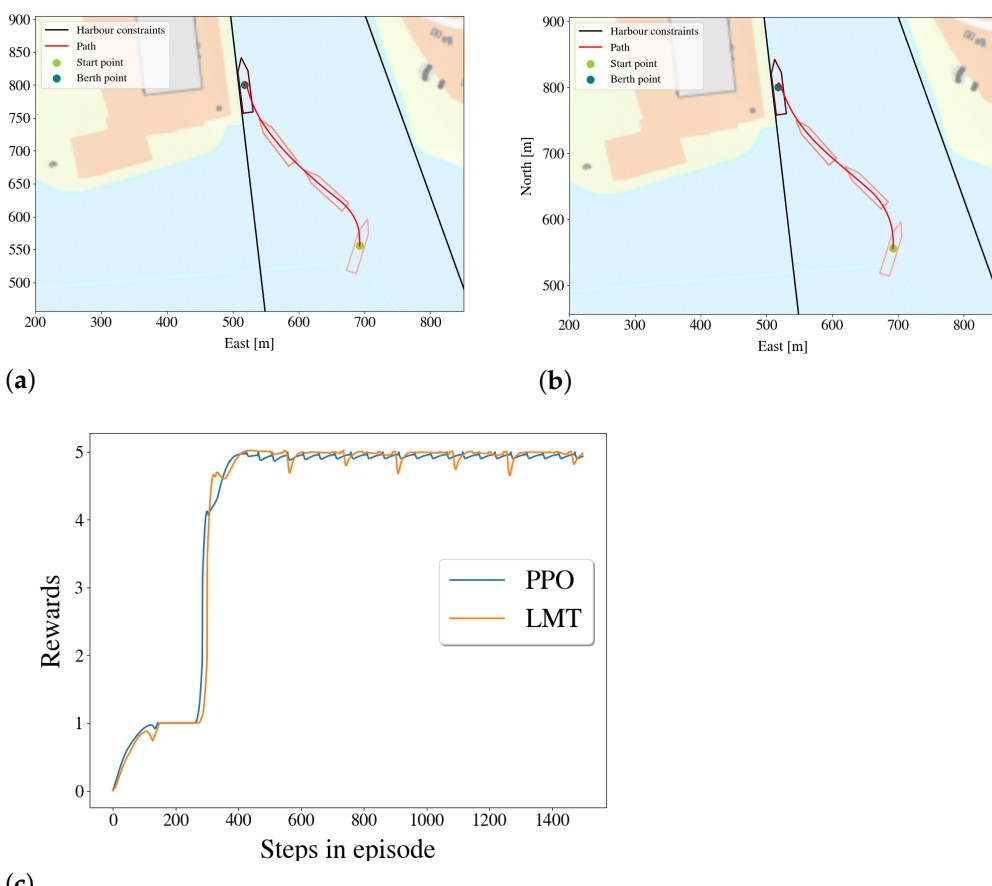

**Figure 11.** (**a**) A successful run by the LMT OFS 312, (**b**) a sucussefull run by the PPO-policy, and (**c**) the LMT's and PPO-policy's rewards for their respective runs with the same starting points.

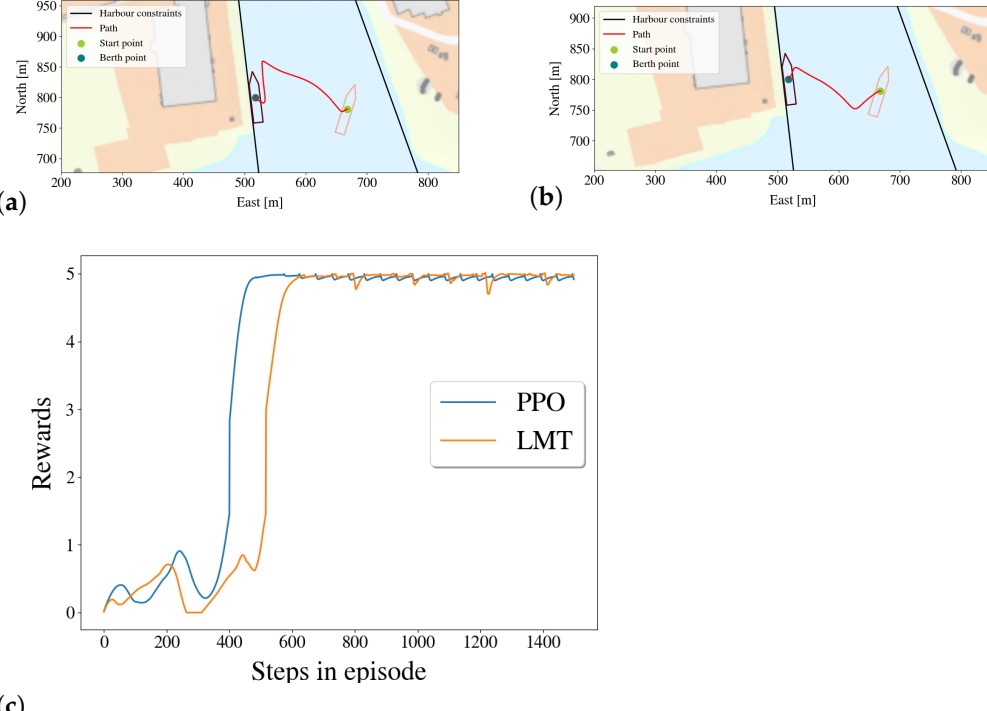

**Figure 12.** (**a**) A successful run by the LMT OFS 312, (**b**) a successful run by the PPO-policy, and (**c**) the LMT's and PPO-policy's rewards for their respective runs with the same starting points.

## 6. Discussion

The main drawback with the building process outlined in Algorithm 1 is that it is a heuristic, greedy method. This means that there is no guarantee of an optimal approximation of the black-box method, nor any guarantee of optimality given a dataset or a given tree structure. One of the main issues with building LMTs in a greedy way is that good splits hidden behind seemingly bad splits will not be found. This has been addressed by adding some randomness to the building process to further explore the solution space. Algorithm 1 can be sensitive to outliers in the dataset since a larger range in the features' values will stretch out the thresholds' grid search. The linear regression may also be affected by outliers. For this reason, alongside the fact that the LMT cannot learn aspects of the black-box model's behavior that are not represented in the dataset, it is important to have a good dataset. For this application, one tree with five linear functions in each leaf node was chosen due to the fact that building five trees is much more computational demanding than building one. As discussed, transparency for both DTs in general and LMTs depends on the size of the tree. If a small enough tree can be made with decent fidelity to the black-box model it is approximating, an assessment between accuracy and interpretability must be made. In this work, all the trees considered are too big to be categorized as simulatable transparent, thus only accuracy should be taken into account when choosing which tree should be the explainer model for the black-box model. If an LMT was to approximate the PPO-policy with adequate accuracy, the LMT may replace the PPO-policy entirely. This is beneficial because then the correctness of the feature attributions would be guaranteed. Introducing feature ordering to the splits improved the accuracy of the trees while decreasing their size, but still, good splits can be hidden behind bad splits which will not be found due to the building process' greedy nature. The ordering of which features can be searched for at different depths of the tree should be done in a way that makes sense in terms of the application, but there may be many orders that could work, so several alternatives should be tested. In this work, the explanations come in the form of feature attributions which are calculated by using the linear function in the activated leaf node. This means that the splits along the path from the root node to the activated leaf node are not taken into account when forming the explanation, even though it clearly is important. Say that a leaf node gives out a constant prediction through the coefficient $w_F + 1$ from Equation (5), then the feature attributions will all be zero, and thus there are no explanations for this region. For this problem, the thruster's force and angle are controlled directly instead of having the vessel be controlled through a total force and torque applied to the vessel. This is desirable because the DRL-agent gets more freedom to learn new strategies, but it provides an additional challenge to the XAI-method because it gets less clear what the DRL-policy is attempting to do because there will be many combinations of forces and angles of the thrusters that equal the same total force and torque. Additionally, $a_1$ and $f_1$ controls the same azimuth thruster, and how each of these actions affects the vessel is heavily dependent on each other. For example, if $f_1$ gives no force, the angle, $a_1$, of the thruster does not affect the vessel in any way. As pointed out by [37], interpretability is not a concept that is easily objectively measured, and how the explanation is communicated to the end-user is of great importance to how well the model will be understood. Thus, the visualizations of the vessel's states, actions, and corresponding feature attributions should be evaluated by the users themselves in terms of how factors such as how efficiently the information is communicated, and how they affect the users' trust towards the system. Additionally, how the trees' structure can be used to form better explanations should be investigated, and a more systematic approach to the reordering of the features used in the splitting should be looked into. To summarize the main points of the discussion:

- There are no guarantees for optimality;
- The LMTs are not small enough to be simulatable transparent;
- The splits in the trees are not used when forming the explanations, even though they are of importance;
- In regions where the LMT makes a constant prediction, no explanations can be made;

- Ordering feature splitting significantly improved both the accuracy and build time of the LMTs because the search process for each split becomes faster, in addition to that the iterative data sampling process becomes unnecessary;
- The two user-adapted visualizations of the explanations should be evaluated by the said end-users.

### 7. Conclusions

There is a clear need for XAI-methods for black-box methods, such as DNNs, to be used in marine robotics in general, but the need is also clear for ASVs specifically. In this work, the preliminary work from [27] was significantly extended through improving the algorithm, more thorough testing of the approximation, and better communication of the feature attributions through user adapted visualizations. The algorithm was improved by introducing ordered feature splitting to the trees, both in terms of more accurate trees and in faster building time. This makes the LMTs capable of tackling more complex problems with higher dimensions. Different users require different types of explanations, as well as different representations of both the information about the ASV and the explanation given by the LMTs. Therefore, two different visualizations were suggested for two different users—the developer and the seafarer/operator. The visualizations of the feature attributions do not serve as a full explanation of the model, but can be used as a step towards understanding, or at least trusting, the model.

**Author Contributions:** V.B.G. and A.M.L. conceived the presented idea. V.B.G. developed the software and performed the simulations. V.B.G. and O.A.A. formalized the overview of the two different end-users and worked out their respective visualizations presented in Section 4.2. V.B.G. wrote the manuscript with support from I.S. and A.M.L. All authors contributed to the final manuscript. All authors have read and agreed to the published version of the manuscript.

**Funding:** This work was supported by the Research Council of Norway through the EXAIGON project, project number 304843.

**Data Availability Statement:** The raw data supporting the conclusions of this article will be made available by the authors, without undue reservation, to any qualified researcher.

**Acknowledgments:** We thank Tim Miller at The University of Melbourne for providing insightful feedback and proofreading of the manuscript.

**Conflicts of Interest:** The authors declare no conflict of interest.

### Appendix A. The Docking Agent

The DRL-agent used in this work was trained in [17]. Details about the DNN, the reward function, and parameters used for the PPO-algorithm are given here. For a more detailed description, the reader is referred to [17]. Due to the task's complexity, it was decided to divide the docking problem into several phases and use the results from the subtasks as a warm start for the full task. The docking problem was divided in the following way:

1. Dynamic positioning involves getting the vessel to a specific point and keeping the vessel there. Here the vessel started in close proximity to the point;
2. Berthing involves getting the vessel to the berthing point and keeping it there, starting in close proximity to the berthing point;
3. Target tracking involves getting the vessel in the vicinity of the berthing point and keeping it there, starting from the outside of the harbor;
4. Distance berthing involves performing berthing from larger distances.

How the different subtasks relate to each other are shown in Figure A1. As can be seen in Figure A1, the reward function for the dynamic positioning task was designed before adding components to the reward function for performing berthing. Distance berthing can be seen as a combination of berthing and target tracking. Dividing the task into subtasks allowed for optimizing the reward function for each subtask, as well as

confirming that the rewards for that subtask make sense. After refining the reward function for the three subtasks dynamic positioning, berthing, and target tracking, the reward function for berthing and target tracking was combined to form the reward function used for the task of performing distance berthing, shown in Equation (3) and repeated here for convenience with more details:

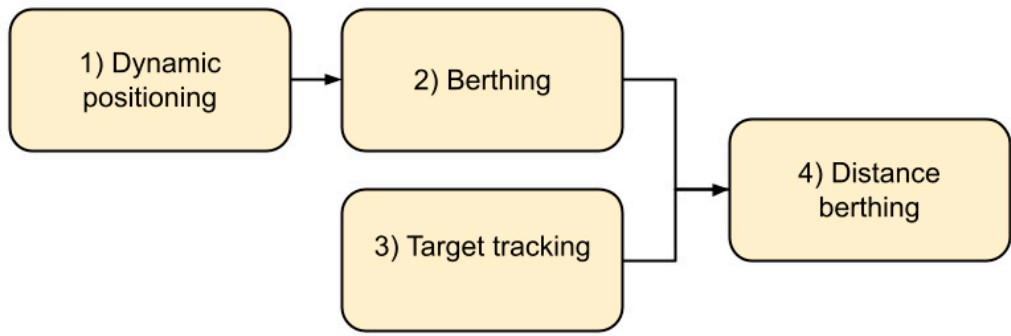

**Figure A1.** The different subtasks of the docking problem in relation to each other.

$$r(\tilde{x}_d, \tilde{y}_d, l, d_{obs}) = r_{d_d} + r_{\tilde{\psi}} + r_{obs} + r_{d_{dot}},\tag{A1}$$

where the different components of the reward function are defined as follows:

$$r_{d_d} = \begin{cases} C_{d_d} e^{\frac{-(d_d^2)^2}{2\sigma_{d_d}^2}}, & \text{if } l = 0 \text{ and } |\tilde{\psi}| < \frac{\pi}{2}, \\ 0, & \text{otherwise.} \end{cases}\tag{A2}$$

where $d_d = \sqrt{\tilde{x}^2 + \tilde{y}^2}$ is the distance from the origin of the vessel to the berthing point.

$$r_{\tilde{\psi}} = \begin{cases} C_{\tilde{\psi}} e^{\frac{-(\tilde{\psi}^2)^2}{2\sigma_{\tilde{\psi}}^2}}, & \text{if } l = 0 \text{ and } r_{d_d} < \frac{C_{dock}}{2} \\ 0, & \text{otherwise.} \end{cases}\tag{A3}$$

$$r_{obs} = \begin{cases} C_{obs} e^{\frac{-(d_{obs}^2)^2}{2\sigma_{d_{obs}}^2}}, & \text{if } l = 0 \text{ and } |\tilde{\psi}| < \frac{\pi}{2} \\ C_{obs,T}, & \text{otherwise.} \end{cases}\tag{A4}$$

$$r_{d_{dot}} = \begin{cases} 0, & \text{if } \dot{d}_d > 0 \text{ and } |\tilde{\psi}| < \frac{\pi}{2} \\ C_{d_{dot}} \dot{d}_d, & \text{if } \dot{d}_d < -1 \\ C_{d_{dot}} \dot{d}_d, & \text{otherwise.} \end{cases}\tag{A5}$$

The parameters for the different reward components in the reward function is given in Table A1.

**Table A1.** Parameters used in the reward function.

| $C_{obs,T}$ | $C_{obs}$ | $C_{d_d}$ | $C_{\tilde{y}}$ | $C_{\dot{d}_d}$ | $\sigma_{obs}$ | $\sigma_{d_d}$ | $\sigma_{\tilde{y}}$ |
|---|---|---|---|---|---|---|---|
| −600 | −2.5 | 2.5 | 2.5 | 1 | 1 | 10 | 0.17 |

The agent was trained using the PPO-algorithm from [28], and the parameters for the PPO-algorithm is given in Table A2. The DNN for both the policy- and value-function was fully connected and had 2 hidden layers of 400 neurons each. The activation function between the two hidden layers was ReLU, while the activation function on the output was

tanh. The size of the network, as well as the parameters for the PPO-algorithm and the reward functions, were found through trial and error.

**Table A2.** Parameters used for the PPO-algorithm.

| | |
|---|---|
| Mini batch size | 20,000 |
| Replay buffer size | $10^6$ |
| Actor learning rate | $3 \times 10^{-4}$ |
| Critic learning rate | $10^{-3}$ |
| Discount rate $\gamma$ | 0.99 |
| Number of epoch updates with minibatch | maximum 8 |
| GAE parameter $\lambda$ | 0.96 |
| Clipping range | 0.2 |

**Appendix B. The Simulated Environment**

The environment consists of two components that need to be simulated, namely the vessel and the harbor. In Appendices B.1 and B.2, how the vessel's dynamics and shape are simulated in [17] is described. In Appendix B.3, how the simulated docking area is defined in [17] is described.

*Appendix B.1. Vessel Dynamics*

The vessel was modeled as a rigid-body mass with no external forces and three degrees of freedom, giving the following two equations of motion:

$$\dot{\eta} = R(\psi)\mathcal{V}, \tag{A6}$$

and

$$\mathbf{M}\dot{\mathcal{V}}_r + \mathbf{D}\mathcal{V}_r = \tau_{control}, \tag{A7}$$

where $\mathbf{M}$ is the rigid-body matrix, and $\mathbf{D}$ is the constant damping matrix. The position vector $\eta = [x, y, \psi]^T \in R^2$ is given by the Cartesian coordinates $(x, y)$ and the yaw angle $\psi$. The velocity vector $\mathcal{V} = [u, v, r]^T \in R^2$ is given by the linear velocities $(u, v)$ and the yaw rate $r$, and $\mathcal{V}_r$ is the vessel's velocity relative to the ocean current. The mapping from the actions directly controlling the thruster angles and forces to the control forces and moments are done in $\tau_{control}$. This mapping was first shown in Section 5.3.3, but is repeated here for convenience:

$$\tau_{control} = T(\mathbf{a})\mathbf{f} = \begin{bmatrix} F_x \\ F_y \\ T \end{bmatrix}, \tag{A8}$$

where $\mathbf{a}$ is the vector containing the control actions for the angles of the thrusters, and $\mathbf{f}$ the control actions for the force of the thrusters. $F_x, F_y$, and $T$ is given by

$$F_x = \sum_{i=1}^{3} f_i cos(a_i), \tag{A9}$$

$$F_y = \sum_{i=1}^{3} f_i sin(a_i), \tag{A10}$$

$$T = \sum_{i=1}^{3} f_i(l_{i_x} sin(a_i) - l_{i_y} cos(a_i)). \tag{A11}$$

The rotation matrix $R(\psi)$ is given by

$$R(\psi) = \begin{bmatrix} cos(\psi) & -sin(\psi) & 0 \\ sin(\psi) & cos(\psi) & 0 \\ 0 & 0 & 1 \end{bmatrix}. \tag{A12}$$

The state of the next time step $t + 1$ is calculated using Euler's method:

$$\eta_{t+1} = \mathbf{R}(\sigma_t)\mathcal{V}_t h + \eta_t, \tag{A13}$$

$$\mathcal{V}_{r,t+1} = (\mathbf{M}^{-1}(\tau_t(\alpha_t, f_t) - \mathbf{D}\mathcal{V}_{r,t}))h + \mathcal{V}_{r,t}), \tag{A14}$$

where $h$ is the size of the time step.

*Appendix B.2. Vessel's Shape*

The vessel's shape is approximated by a pentagon and has the following spatial constraints

$$S_v \in \{o \geq \mathbf{A}_b \mathbf{p}^b - \mathbf{b}_b\}, \tag{A15}$$

where $\mathbf{p}^b = [x^b, y^b]$ are Cartesian coordinates in body frame,

$$\mathbf{A}_v = \begin{bmatrix} -1 & 0 \\ 2.72 & -1 \\ -2.72 & -1 \\ -1 & 0 \\ 0 & -1 \end{bmatrix}, \tag{A16}$$

and

$$\mathbf{b}_v = \begin{bmatrix} -7.7 \\ 41.91 \\ 41.91 \\ 7.7 \\ -41.91 \end{bmatrix}. \tag{A17}$$

For safety measures, the shape was enlarged by 10% to ensure that the vessel does not crash into the quay when docking.

*Appendix B.3. Docking Area*

The docking area is represented by the convex set $S_d$, which is defined as

$$S_d \in \{0 \geq \mathbf{A}_d p^n - \mathbf{b}_d\}, \tag{A18}$$

where $p^n = [x^n, y^n]$ are Cartesian coordinates in the NED-frame,

$$\mathbf{A}_d = \begin{bmatrix} -8.57 & -1 \\ 0 & -1 \\ -0.51 & -1 \\ -2.77 & -1 \\ 0 & -1 \end{bmatrix}. \tag{A19}$$

and

$$\mathbf{b}_d = \begin{bmatrix} 5163.85 \\ 1242.0 \\ 1503.91 \\ 2846.56 \\ 120.0 \end{bmatrix}. \tag{A20}$$

Since both the shape of the vessel and the shape of the docking area are represented by convex sets, the binary variable $l$ stating whether or not the vessel has made contact with the harbor can be defined as

$$l = \begin{cases} 1, & \text{if } \mathbf{A}_s p - \mathbf{b}_s < 0, \forall p \in Vertex(S_d) \leq 0. \\ 0, & \text{otherwise} \end{cases} \tag{A21}$$

The berth rectangle can also be defined as a convex set

$$S_{be} \in \{0 \geq \mathbf{A}_{be}\mathbf{p}^n - \mathbf{b}_{be}\}, \tag{A22}$$

where $\mathbf{p}^n = [x^n, y^n]$ is the Cartesian position in NED-frame. The vertexes of the overlapping area between the berth area and the vessel can be found by using the *Shoelace formula* on the following two inequalities:

$$\mathbf{A}_v p - \mathbf{b}_v < 0, \forall p \in Vertex(S_{be}). \tag{A23}$$

$$\mathbf{A}_{be} p - \mathbf{b}_{be} < 0, \forall p \in Vertex(S_v). \tag{A24}$$

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
