# Peer review of "Explaining a Deep Reinforcement Learning Docking Agent Using Linear Model Trees with User Adapted Visualization"

_jmse, doi:10.3390/jmse9111178_

Round 1
Reviewer 1 Report
Authors address the problem of AI assisted docking of autonomous surface vessels with aim to make AI' s decisions visible and comprehensible to end user where end user may be developer or operator/seafarer.
It is a quality paper with
title that corresponds to the work it presents abstract that summarises well the paper's body
introduction that clearly introduces the state of the art methods related sections that explain used procedures with good scientific soundness results and conclusions that support claims indicated beforehand.
It is of high interest for this journal's readership and broader, it is based on the relevant quality literature and technically correct.
I have no requests for further improvements of this manuscript.
Author Response
Dear reviewer 1,
Thank you for reading the paper and providing your evaluation.
Sincerely,
Vilde Gjærum (on behalf of all authors: Vilde Gjærum, Inga Strümke, Ole-Andreas Alsos, and Anastasios M. Lekkas)
Reviewer 2 Report
This paper provides a linear model tree to approximate deep neural networks. The LMT is applied to understand the black box of how DNN chooses its decision in autonomous surface vessels. The idea of the paper sounds interesting, but there are some issues with the manuscript, which should be considered by the authors.
- Two main contributions are discussed, but they seem insufficient. Overview of background can not be considered as a contribution.
- On page 2, more clear and detailed discussions are required when mentioning the DRL agent of Reference [17].
- Add some discussions about the environment in Table 2. Also, the type of states and action space should be declared in this table. What kind of approximator is applied?
- In the last paragraph of Page 4, describe D_obs and ψ_obs separately. The definition is confusing.
- Choose another character rather than R for penalties.
- The experiments are done in a simulator, while the name and properties of the simulator have never been discussed.
- 400 neurons are selected for each hidden layer of the neural network. What is the purpose of choosing such a specific number? how the authors come put with this value.
- It is mentioned that the agent converges by 6 million interactions. What does the interaction refer to, is it episode numbers or steps in each episode?
- On page 9, it is mentioned that "If the criteria aren’t met after trying all
317 features in the three feature groups, the tree stops growing.", the authors should declare stop growing will cause what kind of drawbacks and advantages, and are the advantages a good contribution to follow. - On page 10, the authors refer to subfigure, a, b, and c of Figure 5, but the subfigures are not numbered within this figure.
- The tables and figures should be referred to in the correct order within the context. For example, the first time that Figure 4 is referenced in the manuscript is after referencing Figure 5, which is inappropriate.
- The caption of Table 4 is insufficient, the authors need to mention more details.
- The last sentence of Subsection 5.1. is incomplete and is not clear. It might need rewording.
- The alignment of all the texts within a table should be the same. In table 6 some texts are aligned in the centre which some are aligned left. Please justify all with the same format.
- In line 2 of the discussion, the first "there" should be removed from "This means there that there is no guarantee of an optimal
483 approximation of the black-box method". - The discussion section is not appropriately organised and it is not clear what are the drawbacks, what are the solutions. Maybe some bullet points could be helpful.
Author Response
Dear Reviewer 2,
Thank you for reading the paper and providing your evaluation. Please see the attached documents.
Sincerely,
Vilde Gjærum (on behalf of all authors: Vilde Gjærum, Inga Strümke, Ole-Andreas Alsos, and Anastasios M. Lekkas)

Round 2
Reviewer 2 Report
All my comments are covered.